# A Detoxification Intervention for Gulf War Illness: A Pilot Randomized Controlled Trial

**DOI:** 10.3390/ijerph16214143

**Published:** 2019-10-28

**Authors:** Kathleen Kerr, Gayle Morse, Donald Graves, Fei Zuo, Alain Lipowicz, David O. Carpenter

**Affiliations:** 1Department of Family and Community Medicine, University of Toronto, Toronto, ON M5G 1V7, Canada; k.kerr@utoronto.ca; 2Department of Psychology, The Sage Colleges, Troy, NY 12180, USA; morseg@sage.edu (G.M.); graved@sage.edu (D.G.); 3Institute for Health and the Environment, University at Albany, Albany, NY 12144, USA; 4Applied Health Research Centre, St. Michael’s Hospital, Toronto, ON M5G 1B1, Canada; zuof@smh.ca; 5Trillium Gift of Life Network, Ministry of Health and Long-Term Care, Toronto, ON M5G 2C9, Canada; alainlipowicz@gmail.com

**Keywords:** Gulf War illness, pesticides, organophosphates, chemical warfare agents, exposure, exposome, Hubbard, sauna, detoxification, nicotinic acid, Veteran’s SF-36

## Abstract

Approximately 30% of the 700,000 US veterans of the 1990–1991 Persian Gulf War developed multiple persistent symptoms called Gulf War illness. While the etiology is uncertain, several toxic exposures including pesticides and chemical warfare agents have shown associations. There is no effective medical treatment. An intervention to enhance detoxification developed by Hubbard has improved quality of life and/or reduced body burdens in other cohorts. We evaluated its feasibility and efficacy in ill Gulf War (GW) veterans in a randomized, waitlist-controlled, pilot study at a community-based rehabilitation facility in the United States. Eligible participants (*n* = 32) were randomly assigned to the intervention (*n* = 22) or a four-week waitlist control (*n* = 10). The daily 4–6 week intervention consisted of exercise, sauna-induced sweating, crystalline nicotinic acid and other supplements. Primary outcomes included recruitment, retention and safety; and efficacy was measured via Veteran’s Short Form-36 (SF-36) quality of life, McGill pain, multidimensional fatigue inventory questionnaires and neuropsychological batteries. Scoring of outcomes was blinded. All 32 completed the trial and 21 completed 3-month follow-up. Mean SF-36 physical component summary score after the intervention was 6.9 (95% CI; −0.3, 14.2) points higher compared to waitlist control and 11 of 16 quality of life, pain and fatigue measures improved, with no serious adverse events. Most improvements were retained after 3 months. The Hubbard regimen was feasible, safe and might offer relief for symptoms of GW illness.

## 1. Introduction

A total of 956,000 military personnel, including about 700,000 from the US, served in the 1990–1991 Persian Gulf War. Within two years following service, approximately 30% of US veterans had developed persistent health effects in a complex multi symptom but variable illness pattern referred to as Gulf War Illness (GWI). Allied countries including UK, Canada, Australia and others also reported increased illness in their Gulf War veterans [1]. Symptoms of GWI include chronic musculoskeletal pain, headaches, fatigue, insomnia, cognitive problems, poor balance, rash, dyspnea, gastrointestinal symptoms and sensitivity to odors [2,3]. 

Many veterans with GWI are significantly disabled. Donta et al. [4] found 42% of veterans with GWI in their study were receiving disability payments and 24% had a pending disability claim. In a 10-year population-based survey follow-up, approximately 30% of all GWI veterans reported physical impairment that limited employment or household work activities [5]. Medical management of these ill veterans has mainly been addressed by control of symptoms, along with integrative approaches targeting oxidative stress with Coenzyme Q10 [6] and carnosine [7], or general supportive measures such as mindfulness [8] and acupuncture [9], all of which showed some promise of benefit. However, no trials have targeted the various GW exposures in terms of attempting to reduce xenobiotic body burden. Three committee reports from 1999, 2008 and 2013 noted urgent need for effective therapies and listed detoxification as being sought by veterans [10] and/or needing investigation [1], the 1999 conference report from the Centers for Disease Control and Prevention (CDC) suggesting a need to “examine the efficacy of a detoxification routine consisting of saunas, stringent exercise, and vitamin therapies” [11].

The symptoms that persisted in these veterans were not found in veterans of the Gulf War who only went as far as Germany [12] or those in other Middle East wars. While the actual ground campaign lasted only four days, there were months of prior buildup and aerial bombing during which time troops reported hearing chemical alarms. During the period January 19–24, 1991, dedicated detection equipment confirmed the presence of sarin and sulfur mustard [13]. Troops were also exposed to multiple pesticides, used to prevent sand fly fever [14], and many took an experimental anti-soman nerve agent medication, pyridostigmine bromide [15]. There were extended oil well fires following Iraq’s surrender, with potential dioxin exposure [16]. There were several other potentially hazardous exposures unique to this brief war [2,17,18], including depleted uranium, multiple vaccines, fuel and exhaust fumes, chemical agent resistant coatings and airborne sand particles. 

With the exception of depleted uranium and polycyclic aromatic hydrocarbons from combustion there have been no internal dose assessments [19,20]. Assessment of exposures has been based on recall questionnaires, job descriptions, zones of deployment [17] and modeling [13,21]. Few of the potential exposures are of a known persistent nature with the exception of possible dioxin/furan exposures as above, and possibly polychlorinated biphenyls (PCBs). A Kuwait oil fire health risk assessment of air and soil concluded a small potential for adverse health impacts from presence of dioxins and furans [22]. Dated sediment core measurements from the northwestern Arabian Gulf identified a 1991 spike in PCBs that may have been due to burning of PCB-containing power transformers during the Iraqi occupation of Kuwait [23]. A study of adipose total PCBs from the Baghdad area of Iraq found levels more than twice the US background, but the source and era of exposure to the PCBs is uncertain [24]. Sulfur mustard, which was detected in fallout from aerial bombing early in the war and which possibly exposed up to 40% of the troops [13], is lipophilic, genotoxic and may be retained in subcutaneous adipose tissue [25]. 

Relationships of non-specific symptoms to complex war-related or other background exposures are very hard to untangle. It is worth recalling the work from the 1970s after animal feed in Michigan was accidently contaminated with polybrominated biphenyls (PBBs), which clearly documented the onset of non-specific neurological symptoms after exposure [26]. In our era, we no longer have unexposed populations to compare to.

While the cause of GWI remains uncertain and is likely multifactorial, many symptoms are related to nervous system dysfunction. The most likely candidate neurotoxic exposures are to acetylcholinesterase inhibitors, which include pyridostigmine bromide, organophosphorus pesticides and organophosphate nerve gases such as sarin and cyclosarin [3,27]. Exposure to high dose organophosphorus pesticides results in acute inhibition of acetylcholinesterase, but chronic low dose effects may include muscle weakness, polyneuropathy and neuropsychiatric symptoms, including fatigue, autonomic dysfunction, depression, mood changes and problems with attention and information processing [28,29]. Survivors of the Tokyo subway attack with sarin gas have exhibited chronic effects on psychomotor performance and vestibulo-cerebellar system function, difficulties in postural balance and psychiatric symptoms [30]. 

Many years have passed since these war-related exposures, and while there are many confounders, the continuing symptoms may be hypothesized to be due to some persistent chemical agent, metabolite, or antigens, permanent damage to the brain and/or immune system, genotoxic or mitochondrial damage or some combination of persistent exposure and tissue damage [31,32,33,34,35]. Rates of post-traumatic stress disorder and other psychiatric conditions were found to be relatively low in Gulf War veterans as per the 2008 report from the Research Advisory Committee [1] on Gulf War Veterans’ Illnesses. This reported also noted that “evidence strongly and consistently indicates” that exposure to pesticides and pyridostigmine bromide pills are causally associated with GWI, adding that low-level exposure to nerve agents, close proximity to oil well fires, receipt of multiple vaccines and effects of combinations of Gulf War exposures could not be ruled out as causes. Due to the consistency of findings relating GWI to neurotoxic exposures, the Committee advised that there be studies of “technologies capable of detecting toxins or secondary metabolites retained for many years following exposure”... or… ” persistent” or “downstream” changes in biochemical processes in relation to past neurotoxicant exposure” [1]. Thus, internal exposure has been considered.

There is accumulating evidence regarding pathophysiology. GWI veterans show both anatomic and immune function differences, mitochondrial DNA damage and dysfunction [32,34]. Chao et al. [35] used MRI to examine brains of Gulf War veterans who were exposed to sarin and cyclosarin in Khamisiyah, Iraq, and reported reduced white and gray matter volume as compared to controls. Engdahl et al. [36] reported altered function in the frontal cortex and cerebellum and Christova et al. [37] have reported subcortical brain atrophy. Abou-Donia et al. [38] and Georgopoulos et al. [39] have both shown altered neuro-immune markers in veterans with GWI.

We are making no claims that veterans with GWI have abnormally high levels of specific toxins, only that the unusual set of GW deployment exposures (along with genetics and other factors) are likely linked with precipitating the illness. Humans are continuously exposed to complex mixtures of xenobiotics, which for these veterans would have occurred prior to, during and following the Gulf war exposures, all contributing to accumulated body burden. As described by Carpenter [40] in 2002 and recently by Sexton [41] “Assessment of cumulative risk … involves evaluation of collective health effects of multiple stressors—as opposed to individual effects of a single stressor; ……… it takes account of background exposures…”. Additionally, Wild [42,43] and others [41] have now recognized the importance of the exposome as a complement to the genome. The exposome is composed of “every exposure to which an individual is subjected from conception to death. Therefore, it requires consideration of both the nature of those exposures and their changes over time” and “Although the risks of developing chronic diseases are attributed to both genetic and environmental factors, 70 to 90% of disease risks are probably due to differences in environments” [44].

This study assumes that if GWI is associated with some toxic exposure(s), an interventional strategy comprising a detoxification approach that could either reduce the burden of the toxins or reduce the toxic effects should be investigated. The specific intervention we are studying, developed in 1978 by Hubbard, is based on known principles which can be synergistically exploited to gradually increase the rates at which the body can both mobilize and excrete lipophilic and other xenobiotics [45,46]. Undertaken daily for four to six weeks, it combines gradually increasing doses of immediate release nicotinic acid, 20 to 30 min of moderate aerobic exercise, two to four hours of intermittent sweating in a low temperature sauna, electrolytes, polyunsaturated oils and multivitamins/minerals. 

The biologic plausibility for this concept is supported by literature on each of the elements of the regimen, described in full detail in the Food and Drug Administration (FDA) Investigational New Drug (IND) Investigator’s Brochure (See S1 Investigators Brochure). The term detoxification would refer here to an applied process, a set of steps systematically taken in combination, with the intention to enhance endogenous systems of biotransformation and elimination, and thereby reduce the body burden of toxic substances. Other examples of detoxification interventions from the field of toxicology include management of acute poisoning using hemodialysis or activated charcoal [47,48], chlordecone toxicity managed with cholestyramine, an anion-exchange resin [49] and ingestion of the non-absorbable dietary lipid, olestra, to reduce body burden of dioxins or PCBs [50]. 

The Hubbard method has been investigated in several studies since the 1980s and has shown promise in reducing body burdens and/or improving symptoms in other exposed populations. Small early trials reported reduction of persistent lipophilic xenobiotics (PBBs, PCBs, DDE) by 25–30% in adipose tissue and serum, with a case report of skin lipid measurements of PCB concentration doubling on the 14th day, suggesting this is one of the elimination pathways enhanced in the regimen [51,52]. Other studies in several populations with differing exposures reported improved IQ [53], better neurocognitive function in firefighters (trails B and block design) [54] and ability to work in residents exposed to Chernobyl radioactivity [55]. Pain, fatigue and SF-36 quality of life scores improved in policemen exposed to illicit methamphetamine laboratory explosions [56]. These policemen had chronic symptoms similar to those experienced by GWI veterans. Respiratory symptoms improved after the Hubbard regimen in first responders at the World Trade Center fires and collapse exposures [57], similar to some GW veterans exposed to oil well fires with respiratory symptoms and disease [58]. While these reports to date suggest that the method may reduce body burdens and improve core symptoms associated with xenobiotic exposures, the studies have limitations, such as having small sample size or being open label in design, leading to potential internal bias. Three were non-randomized controlled studies. No serious safety issues have been reported in studies to date. There have been no studies of populations with remote organophosphate or nerve gas exposure and there have been no feasibility or systematic randomized controlled trials of this intervention in veterans with GWI. The focus of the current work was limited to determining if there is an improvement in health and function and was not designed to study associations between symptoms and body burdens of specific chemicals.

In order to increase options for interventions that might provide remission of symptoms, we aimed to determine feasibility, safety and an estimation of effectiveness of the 4–6 weeks Hubbard intervention compared to a waitlisted group in terms of improvements in health-related quality of life (HRQoL), pain, fatigue and neuropsychological measures in ill GW veterans. We also aimed to determine whether any improvements seen would be maintained for as long as 3 months. The results of the neuropsychological tests are presented in a separate publication. 

## 2. Materials and Methods

### 2.1. Study Design

We performed a pragmatic, parallel-group randomized, controlled pilot study at the Severna Park Health and Wellness Center, an existing commercial rehabilitation center that offered the Hubbard detoxification program to firefighters, veterans and others in Annapolis, Maryland, USA. We contracted with the Center to provide the regimen and for office space for the study coordinator. The Center staff treated trial participants as any other individual in attendance there for the program. The study physician was located a two-minute walk from the sauna facility. The immediate start group completed the intervention over about a 4–6 weeks’ time frame with assessments before and 7 days after completing the program. Those allocated to a waitlist were also tested before and after a 4–6 week wait period. The waitlist group participants then underwent the sauna intervention and had a 7 days post-intervention assessment, and both groups had 3-month follow-ups, which gave us three assessment time points for the intervention and four time points for the waitlist group. The time line for the study is found in Figure 1.

This was a pragmatic trial conducted in an existing outpatient clinical setting, which can indicate whether the intervention works in real life conditions. We selected a waitlist control design, as we thought it unethical to withhold the sauna regimen from ill GW veterans who were seeking solutions [59]. The short time frame of 4–6 weeks helped to reduce chance of other outside events changing the health status of this group; however, we did not expect much change, as minimal changes in symptoms and function over time have been documented in long term follow-up studies of GW veterans [60,61]. This design, allowing the waitlist participants to receive the intervention after 4–6 weeks, presented a limitation however, in that we had no untreated waitlist-control 3 month follow-up data as a comparison for the immediate start group. Another limitation was that with small sample size, even with randomization, the groups could be dissimilar; thus, we used adjustment of baseline scores using analysis of covariance (ANCOVA).

Human participants’ research ethics approval was obtained from all participating institutions including Chesapeake Institutional Review Board (IRB), Columbia, Maryland (Pro00007192) after delegation to this IRB by the sponsoring institution, the University at Albany; the Sage Colleges, Troy, New York; Women’s College Hospital affiliated with the University of Toronto, Toronto, Ontario, Canada; and the Human Research Protection Office (HRPO) of the Department of Defense, U.S. Army Medical Research and Development Command. We obtained Food and Drug Administration Investigational New Drug approval for crystalline niacin + heat and exercise (FDA IND # 118015). All participants provided written informed consent. The trial was registered August 2012 at ClinicalTrials.gov: NCT01672710. An independent research monitor had oversight for the safety of the research. This study is reported according to the Consolidated Standards of Reporting Trials CONSORT guidelines.

### 2.2. Participants

Participants were a self-selected sample of Persian GW veterans, defined as having been deployed to the Persian Gulf between August 1990 and July 1991 and found eligible if they met the Kansas case criteria for GWI, which requires moderately severe and/or multiple symptoms in at least three of six symptom domains that first became a problem during or after the 1990–1991 Gulf War [62]. These domains include fatigue, pain, neurological/cognitive/mood, skin, gastrointestinal and respiratory. Each symptom scores 0–3 and to define a case of GWI requires a total score of 2 or greater in at least three of six symptom domains. Although there is as of yet no validated case definition for GWI, in 2014, the Institute of Medicine (IOM) recommended this case definition as best reflecting the symptom complexes of GWI [63]. Per the Kansas criteria, veterans were excluded if they had been diagnosed by a physician with serious chronic conditions, such as cancer, heart disease, liver disease, multiple sclerosis not associated with GW service but involving diverse symptoms, e.g., fatigue, cognitive problems, or pain, similar to those affecting GWI veterans, or conditions that might interfere with a veteran’s ability to report symptoms, e.g., serious psychiatric conditions such as bipolar disease, schizophrenia or a history of hospitalization in the past two years for depression, alcohol, drug dependence or post-traumatic stress disorder.

There were no exclusions for race, age, gender or ethnicity, assuming literacy in English. Program specific exclusions included GWI veterans who were medically not advised to temporarily discontinue certain medications (e.g., anti-hyperlipidemics, antidepressants, analgesics and anti-inflammatories) for the period of the regimen, as taking these could compete with biotransformation of xenobiotics hypothetically mobilized from stores, or could reduce participants’ awareness of changes occurring due to the regimen. Veterans were also excluded if cytochrome p450 inhibitor drugs were recently discontinued (e.g., selective serotonin reuptake inhibitors) and the wash-out period did not span at least five half-lives (per FDA IND recommendation). We made a change in the eligibility criteria after we began recruitment so as to not exclude veterans who had been diagnosed with diabetes later than the onset of their GWI, as long as the diabetes was not severe and was well controlled, or GWI veterans who had mildly elevated liver enzymes (less than three times normal). 

### 2.3. Recruitment and Enrollment

We recruited via electronic media, posters and brochures placed at veterans’ organization sites with great assistance from several veterans’ organizations, as well as media reports and veterans’ word of mouth. Prospective participants contacted the study coordinator for a telephone interview via a scripted prescreening questionnaire, with the broader Centers for Disease Control (CDC) GWI case criteria, which requires one or more moderate to severe symptoms in two of three domains (fatigue, musculoskeletal pain and mood/cognition) that began during or shortly after the war and lasted for at least 6 months [64]. If they met the pre-screening criteria and had no apparent medical exclusions, we invited them to visit the Annapolis facility, complete the written informed consent process and then complete the Kansas Gulf War Veterans Health Project Questionnaire [62], which includes questions on Gulf War exposures and deployment characteristics and 30 questions to elicit the symptom profile for determination of eligibility per Kansas case status. They then had a standardized medical intake examination from the study physician to assess for additional program-specific exclusions including pregnancy, gout, sickle cell trait, being wheelchair bound, having open infectious lesions and having specific illnesses including anemia, liver disease, kidney disease and heart disease other than treated hypertension. The medical intake questionnaire also included questions regarding history of past drug or alcohol problems, head injury and lifetime occupational or other exposures. Blood and urine samples obtained for eligibility and for safety monitoring were analyzed by Quest Diagnostics, Annapolis, MD, for chem-20, lipid profile, complete blood count, thyroid function tests and other tests the physician felt were necessary such as urinalysis and electrocardiogram. The flow of activities in this study is shown in Figure 2.

With explicit signed consent, we obtained an additional blood sample which was centrifuged and the serum shipped and stored frozen at −80 °C at the University at Albany for later analysis of concentrations of persistent organic pollutants, if and when funding becomes available. 

### 2.4. Randomization and Blinding

The University of Albany statistician, who had no contact with the participants, provided a computer-generated simple random allocation sequence of 60 to the study coordinator who created a computer-generated list of unique participant ID numbers, kept in sequentially numbered, opaque, sealed envelopes, which were opened only after a participant had been found eligible. As it is not possible to mask an intensive sauna-based detoxification program, we could not blind participants or staff at the facility to allocation. We also could not blind the facility staff to participants who were GW veterans versus other non-study individuals (e.g., firefighters) attending the sauna facility. All outcome assessments used only ID numbers, which were securely sent to either the University of Toronto (quality of life, symptoms) or Sage Colleges (psychological and cognitive test scores) for scoring and data entry. The data entry research assistants and statistician were blind to the allocation of the ID numbers, but could possibly guess some allocations, since there was potentially an extra set of outcome measures in the control group.

### 2.5. Intervention

The regimen was administered daily seven days per week. Supervisors were in attendance throughout each session and a senior supervisor reviewed each day’s reports and provided written guidance for the next day. The staff was trained according to a detailed manual covering all aspects of the Hubbard method. They were also trained in first aid and in preventing potential problems from use of a sauna or exercise equipment, e.g., heat stress, dehydration, over-hydration, fainting and injury, with special regard to any accommodations required due to symptoms of GWI. Each daily session began with a brief interview to document recent symptoms, hours slept, food eaten, any vitamins or medications taken since the previous day, weight, blood pressure and heart rate. Doses of supplements administered that day were documented on the form. At the end of the day interview, the supervisor noted down what had occurred during the program, weight, blood pressure and heart rate. In addition, the participants themselves documented minutes exercised, hours in and out of the sauna, salt or potassium supplement doses taken, new symptoms that day, realizations during the session and any questions they had. 

Each session started with drinking the specific dose of powdered crystalline niacin (immediate release) dissolved in a glass of water, followed by 20–30 min on a treadmill, exercise bike or elliptical machine at a moderate aerobic level as tolerated. Often the expected niacin flush would begin during the exercise time frame. The next two to four hours were spent in the low temperature (60–80 °C), well-ventilated, Finnish sauna with cool-off breaks, showers, fluids, electrolytes and food as needed, overseen by the supervisor. The niacin flush and any other symptoms would peak during the sauna phase and then dissipate. The other vitamin/mineral supplements were administered throughout the day with glasses of calcium-magnesium (Cal-Mag) drink. The cold-pressed polyunsaturated blended oil and lecithin were taken after the sauna session. The dosages of supplements increased as the program progressed, through five stages (Appendix A) but could also be adjusted as recommended by the study physician and depending on tolerances. There were no changes from the participant’s normal diet apart from requirement to eat some vegetables and weight change was not expected.

Supervisors explained to participants the expected reactions to niacin including a hot, red skin flush and itch, which could last up to an hour or longer, could cause emotional changes or difficulty with concentration and could possibly cause a taste or odor suggestive of some chemical to which the participant had been previously exposed. A diminished reaction to a niacin dose on a given day signaled an increase in dose the next day. After up to four or more weeks, when further increasing the daily niacin dose caused no flush or mental/emotional reactions and the participant originated a feeling of wellbeing, the regimen was considered to be complete. This most often occurred at a niacin dose of 5000 mg for the last few days 

We purchased the nutritional supplements from a single supplier, Dr. Price’s Vitamins, Los Angeles (Appendix A) whose raw materials were manufactured by Bactolac Pharmaceuticals, Hauppauge, New York, USA, a facility certified as having passed a Good Manufacturing Practices and Food Safety Systems Audit. Consistency and potency were independently verified by additional testing and documented with Certificates of Analysis for each lot. Supplements were either stored in dark cabinets or refrigerated at the facility and counted and assembled for use for each participant daily. 

### 2.6. Waitlist Control

All participants allocated to the waitlist control group were required to complete baseline tests, go home and then return to the facility after 4–6 weeks to redo the assessments. They were continued during this time on treatment as usual, taking any previously prescribed medications such as for hypertension, diabetes, hypothyroidism etc. Participants from out of town were assisted with local housing during the waitlist period if the travel was too far.

### 2.7. Safety Aspects

We received an Investigational New Drug approval to proceed from the FDA for a “multi-component intervention that includes exercise, daily sauna and dietary supplements including vitamins A, B complex, plain crystalline niacin, C, D and E”. All of the regimen components, heat stress via sauna, exercise and supplements, are in common use, readily available to veterans and are generally regarded as safe, with certain common-sense precautions. A proportion of veterans regularly consume supplements [65]. The use of supplements in this protocol was short term, approximately 30 days, with the highest doses for only a few days. Regular Finnish sauna bathing is associated with reduced risk of fatal cardiovascular events and all-cause mortality [66]. At least 150 min a week of moderate aerobic activity is routine medical advice [67]. Heat stress during the Hubbard regimen does carry some risk of fluid and electrolyte disturbance, which was noted in a case report of dilutional hyponatremia, due to improper supervision of water intake, published in 1977 [68]. We mitigated and/or prevented this by daily monitoring of blood pressure, weight, fluid and electrolyte intake and symptoms by trained supervisors who guarded against over-hydration. The staff was American Red Cross First Aid and CPR certified. The study coordinator also made daily observations of participants and the study physician assessed medical concerns as needed by referral from the sauna supervisor or study coordinator and did laboratory assessments, if needed. If symptoms of heat illness were to occur, the study physician was to be notified and would consider transfer to the nearest medical treatment facility, if necessary. Any unanticipated adverse events were reported on standard US Army Human Research Protections Office (HRPO) adverse event report forms including date of onset, relatedness to treatment and resolution. We had an independent research monitor who was responsible for overseeing the safety of the research and for reporting findings if indicated to the IRB and HRPO. The research monitor and principal investigator independently reviewed each unanticipated adverse event to determine if further action was necessary.

### 2.8. Outcome Measures

#### 2.8.1. Feasibility

We assessed feasibility of implementing and studying the intervention in this veteran population via a log of accrual rates of those approached, assessed for eligibility, enrolled, randomized and retained through phases of the study, which also reflected adherence and acceptability. 

#### 2.8.2. Safety

We assessed safety by daily reports maintained by the participant and the supervisor, structured medical examinations, any additional medical reports and laboratory results of routine blood tests: glucose, lipids, uric acid, renal and liver function, complete blood counts and thyroid function. 

#### 2.8.3. Heath Related Quality of Life and Function

We assessed HRQoL via the Veterans RAND Corporation 36-Item Health Survey (VR-36), a self-report paper and pencil questionnaire. The VR-36 includes eight subscales, with five physical health domains: physical function, bodily pain, general health, role-physical and vitality; and three mental health domains: social functioning, role emotional and mental health. The physical component summary (PCS) score was chosen as a primary outcome, because it is widely used in assessing the health effects of chronic illness [69], and has been used in other trials with GWI [4,9,70]. 

The PCS score integrates the domains, with greater weight given to physical health domains, while the mental-component summary (MCS) score gives greater weight to mental health domains. PCS and MCS scores range from 0 to 100, and are standardized to the US population with a mean of 50 and standard deviation of 10, with higher scores associated with better health. Male and female veterans have clinically comparable scores [71]. We pre-specified that a minimal clinically important difference in PCS scores would be an increase of 7 points based on the reasoning of Donta et al. [4] in a GWI trial who noted that changes of this magnitude had previously been shown to be clinically relevant in studies of chronic illnesses. However, others have estimated that an increase of 3–5 points is also clinically meaningful [72]. For the vitality subscale a 7–11 point increase was cited from a systematic review [73]. Of special interest in our study were the vitality, role-physical and social functioning subscales of the VR-36, which have been found to be sensitive and specific indicators of disability in patients with chronic fatigue syndrome [74]. 

#### 2.8.4. Symptoms and Case Status

We assessed change in participants’ pain symptoms with the revised short-form McGill Pain questionnaire (SF-MPQ-2) [75], which uses recall during the past week of 22 descriptors in four subscales: continuous pain, intermittent pain, neuropathic pain, affective descriptors measured from 0-10. The total pain score is the mean of the 22-item scores. The SF-MPQ-2 has been widely used, including for assessing pain in many chronic conditions and in US veterans [76], and has established reliability and validity [77], including test-retest reliability [78]. A minimally important difference would be expected at a decrease of 2 points [73,79]. 

We assessed changes in fatigue symptoms with the Multidimensional Fatigue Inventory (MFI), composed of five subscales: general fatigue, physical fatigue, reduced activity, mental fatigue and reduced motivation [80]. Scores for each subscale range from 4 (absence of fatigue) to 20 (maximum fatigue). The MFI has been found reliable and valid to assess fatigue in some populations [81], but had questionable internal consistency for general and physical fatigue subscales with chronic fatigue syndrome patients and ceiling effects on the general fatigue subscale [82,83]. The MFI has been used in other studies of GWI [4,84,85,86]. Estimates for minimal clinically important differences for MFI subscales ranged from 1.4 to 2.4 in a radiotherapy study [87].

We used the section of the Kansas Gulf War Veterans Health Project Questionnaire, which covers 30 symptom items in six domains, both to track overall symptom burden and to assess whether the participants continued to meet the Kansas case criteria over time, with a minimum score of at least 3 out of six criteria. We did not enquire as to whether participants had continued to exercise or take supplements after completing the regimen which could have influenced their health status at the 3-month follow-up.

### 2.9. Statistical Methods

As this is a pilot study, a formal sample size calculation was not required [88], but we estimated that a sample size of 50 participants, with 25 in both the treatment and wait-list control, would be adequate to assess overall feasibility of recruitment and to estimate if there appeared to be clinically important improvement of 7 points on the VR-36 Physical Component Summary and improvements in other health measures. Due to time and budget constraints, it was necessary one year into the study to get HRPO approval to reduce our planned sample size to 30. 

We reported feasibility and safety outcomes descriptively as frequencies or percentages. We summarized the descriptive data of symptoms and quality of life outcomes with means and SDs for continuous outcomes and then constructed a model to examine the treatment effect on the VR-36 domains and other symptom measures at 7-days post treatment or waitlist (4–6 weeks time point), while controlling for baseline, using an analysis of covariance (ANCOVA). We did a complete case, intention to treat analysis. Similarly, we also performed an ANCOVA to assess any difference between the immediate sauna group and waitlisted group at 3 months, as the control group had begun their intervention 6 weeks later than the intervention group. For the control group the second baseline at the end of the 4–6 weeks waitlist was used. To assess stability of any improvements, a mixed effects model compared groups from baseline to the 3-month follow-up. Analyses were performed using R language version 3.5.1 for statistical computing (R Foundation for Statistical Computing, Vienna, Austria). 

## 3. Results

### 3.1. Recruitment. and Participant Flow

Recruitment began November 18, 2013 with a target of 50 eligible veterans accrual to be completed in a 12 to 18-month period. With an accrual of only eight veterans entering the study by December 2014, a one-year extension was granted and the target was revised to 30, approved by HRPO, U.S. Army and the IRB. In the second year, accrual rate improved and a further 24 eligible veterans began the study by August 21, 2015, with a total of 32 successfully recruited and retained, achieving the target. Recruitment was closed in August 2015 and the final follow-up completed by the end of October 2015 at which time no remaining funding was available. The CONSORT flow diagram of the study was shown in the Methods section.

### 3.2. Baseline Characteristics

Table 1 shows the baseline demographic and clinical characteristics for each group. Of the total 32 participants who underwent the study, there were 22 in the immediate start group and 10 in the waitlist. We were unable to achieve the intended 1:1 randomization due to our small sample size which increased the probability of chance imbalance [89,90]. Twenty-two participants were male and 10 were female, between the ages of 43 and 70, mean age 51 ± 6.5 and 26 (81%) were Caucasian, 19 (59%) were employed fulltime, three (9%) working part time or retired and 10 (31%) were disabled. Participants had an average body mass index (BMI) of 32 ± 6 kg/m^2^ with a range 25–49 kg/m^2^. Of all participants, 27 (84%) never smoked, five (16%) had diabetes, 17 (53%) remained on prescribed medications, 12 (38%) had stopped taking antidepressants and 10 (31%) had stopped taking analgesics. The two groups were similar in regards to age, sex and BMI, VR-36 MCS scores, diagnosis of diabetes, and staying on medications. Besides the unequal allocation ratio, there were some important differences between the two groups. Over 40% of the immediate intervention participants were disabled while the waitlist controls were all working fulltime. The mean baseline VR-36 PCS scores were 30.3 ± 8.9 and 36.7 ± 10.5 for intervention and waitlist groups, respectively. There were better pain scores in the waitlist controls. 

The PCS and MCS and individual domain scores in our study were very similar to scores in other studies of GWVs [4,64,91] indicating that our sample was likely representative of this population. The vitality domain baseline scores were low, consistent with disability.

We received 221 contacts, but 45 were inquiries from persons who did not complete pre-screening, leaving 176 (Figure 2) who were pre-screened by the study coordinator, generally by telephone. Of these, 67 (38.1%) were ineligible, including 16 (9.1%) who were veterans but not of the first Gulf War, 45 (25.6%) who had medical exclusions such as cancer, recent psychiatric admission or heart disease, and six (3.4%) who could not withdraw from medications such as antidepressants or analgesics. This gave us 109 potentially qualified veterans (62% of total pre-screened) who were invited to an on-site (Annapolis, MD) formal eligibility assessment. Sixty-three (53.3% of those invited) declined to attend, with 33 (18.8%) of these for unknown reasons and 30 (17.6%)] who were interested but unable to participate due to job conflicts, insufficient time or money or travel barriers, as the program required attendance several hours daily for 4–6 weeks. Forty-six veterans (41.3%) attended on-site for consent and final eligibility assessment. Following consent, seven withdrew before starting final eligibility Kansas questionnaire, including four who were unwilling to be assigned to the control group and wait 4 more weeks, two because of lack of their physician’s support, and one for unknown reasons. We randomized a total of 39 persons, 26 in the immediate start group and 13 in the waitlist, which was 22% of the total of 176 veterans prescreened and 36% of the 109 potentially eligible. During the baseline assessment steps, which included medical examination, one in the waitlist group withdrew because of lack of time and three dropped from the intervention group, one of whom was found ineligible as they were not a veteran of the Gulf War. The result was 35 who did the baseline assessments, then one in each group withdrew and one control was found medically ineligible due to having sickle cell trait. The immediate start group included 22 and the waitlist total was 10, an attrition rate after randomization of 18%. 

Overall, the required time and travel commitment affected the success of recruitment, with 30 prequalified and interested but who would not travel to Annapolis, four who withdrew consent after signing the form due to concerns regarding the time commitment of possible allocation to the waitlist control and another after being allocated to the waitlist control; thus, altogether, 35 or 32% of the potentially qualified veterans were deterred due to the time and travel commitment or unwillingness to be assigned to the control group. Many veterans contacting the program were from out of the Annapolis, Baltimore, Washington area. Overall, 44 of 64 (68.8%) potentially qualified veterans who did not attend for the on-site visit and eligibility assessment were from distant states. However, of those who did enroll, over 50% were from distant states including Texas, Louisiana, Colorado, Tennessee, Ohio, Washington State, New York, Vermont, Michigan and Oklahoma. Several veterans requested assistance for travel costs and lodgings, without which they would not have been able to enter the study. Fortunately, veteran groups, other veterans who had done the program and Maryland Veterans Healthcare assisted with finding housing. Charitable groups including the Evangelical Presbyterian Church of Annapolis offered support with lodging, some meals and transport while support for some travel expenses was provided by the Heroes Health Fund.

We assessed retention in two phases: first, the rates of completion in groups allocated to immediate sauna or waitlist. All immediate start participants (*n* = 22) completed the intervention and the 7-day follow-up measures, for 100% retention, (although this included one participant who stopped after 34 days due to gastrointestinal symptoms but was considered to have satisfactorily completed most of the sauna program and who provided all outcome data). All waitlisted controls (*n* = 10) completed their follow-up measurements, for 100% retention. Controls (*n* = 10) then entered the sauna intervention and one withdrew after eight days due to gastrointestinal symptoms and did not complete any further outcome assessments, for 90% retention in this delayed arm. The overall total was 31 of 32 retained, 96.9%, indicating excellent overall acceptability and tolerability of the sauna, exercise, vitamins etc. Adherence to the protocol and patient acceptability also were assessed by review of standardized daily reports completed by each veteran plus free text comments and stated gains after final completion of the sauna regimen. 

The sauna intervention completion time was 25.7 ± 6.7 days, with a range of 16–38 days for all participants (*n* = 31) who were retained, excluding one who withdrew after 8 days due to an adverse effect, and thus was not included in this calculation.

The second phase of retention was defined as returning for the 3-month follow-up. A total of 21 of the 31 enrolled and completed participants (68%) attended. We were not able to schedule five participants (four in the immediate start group and one in the waitlist start group) only because our study termination date was before they reached their 3-month post-intervention. The remaining five losses were for unknown reasons in four and one had died at home of an unrelated cause, as ascertained by the independent research monitor, 6 weeks after completing the intervention. Of note, the 3-month follow-up required an onsite visit for redoing all the questionnaires, neurocognitive test battery, a medical examination and blood draw, which for some participants required cross country travel.

### 3.3. Quality of Life and Symptom Outcomes

Between-group differences in the mean changes after the sauna intervention or treatment as usual waitlist period by ANCOVA, adjusted for baseline, are presented in Table 2. The VR-36 PCS adjusted difference was 6.9 (95% CI; −0.3, 14.2; *p* = 0.06), favoring clinically important improvements in the sauna treatment group. The VR-36 MCS adjusted difference was 9.5 points (95% CI; 3.1, 15.8; *p* = 0.003), favoring the sauna treatment group. VR-36 subscales of role-physical, bodily pain, general health, vitality and mental health scores were all improved in the sauna intervention group compared to the waitlist. An exception was the physical functioning subdomain, which showed no discernable between-group difference. There was a large estimated between-group difference of 31.2 (95% CI; 15.6, 46.9; *p* <0.001) seen for the vitality subscale, which is known to be low in individuals with fatigue and GWI. A vitality subscale score of <35 is considered a substantial reduction in function and for our participants the score at baseline was 23.3 (±23.1) for the intervention group rising to 53.9 (±25.2) after the intervention. Additionally, the role-physical between group difference was large, at 27.6 (95% CI; 6.9, 48.3; *p* < 0.009).

Pain was assessed using the VR-36 bodily pain subscale score and the McGill total pain score. The adjusted difference in the VR-36 bodily pain subscale score was an increase of 26.4 (95% CI; 8.5, 44.4; *p* < 0.004) and the adjusted difference in the McGill total pain score was a decrease of −1.1 (95% CI; −2.0, −0.2; *p* = 0.02). Both represented clinically relevant improvements towards reduced pain. For fatigue, all of the five MFI scores improved. Regarding GWI case status, as expected at baseline, all 32 participants met the Kansas case criteria, but of interest, following the intervention 11 of 22 immediate intervention participant’s scores improved; thus, the 50% no longer met the Kansas criteria for Gulf War Illness. After waiting 4 weeks on the wait-list 2/10 controls or 20% also no longer met criteria.

### 3.4. Three-Month Follow-Up Outcomes

#### 3.4.1. Between-Group Difference’s at 3 Months

At the 3 month mark both groups had received the intervention, the only difference being a 4–6 week delayed start for the control group. The mean change in outcomes, from baseline to 3 months post intervention within groups and differences adjusted for baseline with ANCOVA between groups at 3 months is found in Appendix A. There were no discernable between-group differences at the final follow-up, indicating that for receiving the intervention in the immediate or delayed group, the improvements were the same; except for the McGill total pain score, the immediate intervention improved more than the control group. 

#### 3.4.2. Stability at 3 Months

Stability of improvements in the measures taken after the intervention to 3 months were analyzed with linear mixed effects models included the following variables: treatment group, visit and treatment-by-visit interaction. The outcome variable for each model included measurements at baseline, week 6 and month 3 as in Table 3. The improvements found at the 6-week mark for both groups appeared stable at 3 months. For instance, for the waitlisted group PCS, the magnitude of change from the second baseline just prior to beginning the sauna to week 6 post sauna follow-up was 5.8 (95% CI; 0.46, 11.1), whereas the magnitude of change from week 6 to month 3 was −1.3 ((95% CI; −8.21, 5.66). This suggests that the waitlisted group on average experienced an increase in PCS from baseline to post sauna (week 6), and this effect was sustained from week 6 to month 3. Similarly, for the immediate intervention group, the magnitude of change from baseline to post sauna (week 6) was 7.8 ((95% CI; 4.18, 11.45), and the magnitude of change from week 6 to month 3 was −1.3 (95% CI; −5.34, 2.72). This suggests that the intervention group on average experienced an increase in PCS from baseline to week 6, and this effect was sustained from week 6 to month 3. We also found large post-intervention increases in the VR-36 vitality sub-scores for both groups that were maintained or increased at 3 months. Of note, 29% of the 21 participants who completed the 3-month follow-up still no longer met Kansas case criteria for GWI.

### 3.5. Laboratory Results

We did not detect any concerning patterns of changes in the clinical chemistry or hematology measures taken at each stage of the program (see Appendix A). Blood sugar was monitored for potential increases in diabetics treated with niacin. There were five diabetics (three in the intervention group and two in the control group) who had developed diabetes type 2 after they had developed Gulf War Illness. One in the waitlist was not well controlled, with a baseline fasting glucose of 9.4 mmol/L (170 mg/dL), an increase to 10.4 mmol/L (187 mg/dL) after the 4-week wait list with no niacin and then a slight increase to 10.7 mmol/L (192 mg/dL) after the sauna regimen. The intervention group went from 7.8 mmol/L (141 mg/dL) to 7.3 mmol/L (132 mg/dL) post-sauna, while another went from 7.0 mmol/L (126 mg/dL) to 7.9 mmol/L (142 mg/dL) post sauna. Thus, blood sugar changes in diabetics were small. Mean baseline alanine aminotransferase (ALT) stayed about the same from 30 ± 23 U/L to 31 ± 29 U/L. In one participant, the ALT was mildly elevated at 81 U/L baseline and rose to 160 U/L after the sauna, still in a mildly elevated range. There was no 3-month follow-up data for this participant. Mean gamma-glutamyl transpeptidase (GGT) for all participants slightly decreased from baseline of 35 ± 33 U/L to 28 ± 19 after the regimen, and remained lower at the 3 month follow-up. This is of interest, as serum GGT, even within the reference range, may be a marker of exposure to certain environmental pollutants [92]. We saw a slight decrease in mean low density lipoprotein (LDL) cholesterol from baseline 2.9 ± 0.7 mmol/L (112 ± 27 mg/dL) to 2.7 ± 0.6 (104 ± 23 mg/dL) and an increase in high density lipoprotein (HDL) cholesterol from baseline 1.3 ± 0.3 (50 ± 12 mg/dL) to 1.4 ± 0.4 mmol/L (54 ± 15 mg/dL) after the regimen, as expected from intake of niacin. Uric acid can be increased by niacin and veterans with a history of gout were excluded from the study. Mean uric acid values did not change; however, two participants were out of reference range 1 week after the regimen, with values of 512 and 535 nmol/L (8.6 and 9.0 mg/dL), but had no symptoms of gout. Renal function based on mean estimated glomerular filtration rate (eGFR) improved slightly from baseline 87 ± 14 to 90 ± 15 mL/min/1.73 m^2^ at 1 week follow-up, of possible interest as sweating alone has been used to reduce uremia [93]. Mean electrolytes did not change, except one participant who had Crohn’s disease developed moderate hypokalemia described below. There was a small increase in mean TSH from 1.99 mIU/L at baseline to 2.44 mIU/L at post regimen and back to baseline 2.01 mIU/L at the 3-month follow-up. Three participants had marginally high TSH at baseline (6.54, 6.43, 6.66 mIU/L). Two of these had increases in their TSH to 8.52, and 8.13 while the other one normalized to 1.90 mIU/L. The levels of free T4 remained normal, except for one whose free T4 fell to 9 pmol/L (0.7 ng/dL) after the regimen and was diagnosed at that point with hypothyroidism. Mean complete blood counts were stable apart from one participant at 3-month follow-up who had elevated neutrophil count, and was noted to be currently on an antibiotic for a skin infection. 

### 3.6. Adverse Events

All recorded expected discomforts and side effects of the sauna regimen are shown in Appendix A, and all unanticipated adverse events are shown in Appendix A. In summary, the expected discomforts and side effects of the sauna regimen, such as flushing and itching symptoms from crystalline niacin, were very common. Unanticipated adverse event report forms were submitted for five participants, but none were serious. These events included a probably unrelated sinus infection, which resolved, and non-cardiac chest pain evaluated in the emergency department that resolved. Probably related events included an exacerbation of irritable bowel syndrome, which resulted in the participant withdrawing from the study after eight days of the sauna regimen. One participant needed adjustments of anti-hypertensive medication dosages by the study physician. One event of note was a participant who experienced a 90-second episode of pre-syncope which resolved, an event that was not unexpected from heat/cooling effects, especially as GWI can be associated with underlying autonomic dysfunction [94]. The sauna staff was prepared for this possible aspect and managed it well. There was one participant with a history of bowel resection for Crohn’s disease who did not tolerate the supplements, developed diarrhea for a few days and on the 23rd day of the sauna regimen developed fatigue, weakness and headache. He was referred to a hospital with a moderate hypokalemia (serum potassium of 2.8 mmol/L, range 3.2–5.0 mmol/L), treated with hydration and potassium in the ER and prescribed oral loperamide and potassium chloride. The hypokalemia could probably have been corrected with oral potassium chloride but hospital assessment was advised as a precaution. The symptoms resolved, but diarrhea recurred on day 34 and the participant’s sauna regimen was at that point concluded. 

## 4. Discussion

### 4.1. Feasibility of the Intervention and Study Methods

We found it was feasible to both recruit and provide this intervention, although there were some recruitment difficulties, not unlike other studies of GWI, which is a special population [7,95]. Our final study starts were 29% of potentially eligible GW veterans. The weeks of required daily attendance, and potential need for travel and accommodations away from home were significant barriers, as about half of those enrolled were not from the local area and had to be willing to stay in the Annapolis area for the study. Word of mouth, often via electronic media, seemed the most effective. After an initial slow accrual, and a one-year extension, at the close of recruitment we were receiving increased rates of requests for information regarding the trial. There was 100% attendance and retention in the trial with all immediate or waitlist group participants completing assessments. Of the delayed start sauna, 97% completed it and at 3-months, 68% were retained.

### 4.2. Changes in Health Measures Outcomes

There were improvements in most of the HRQoL and symptom measures post-intervention compared to waitlist, sustained at 3 months. Notwithstanding the small sample size, these results suggest clinically relevant benefit of application of the Hubbard detoxification procedure for veterans with Gulf War illness. The improvements in five of eight VR-36 subscales, namely, role-physical, bodily pain, general health, vitality and mental health, are relevant, as there were large absolute differences shown in these scales between GW and non-GW military personnel in the Iowa Persian Gulf study [96]. The large post-intervention increase in the VR-36 vitality sub-score was similar to a large increase seen in another study of the Hubbard sauna regimen [56]. We obtained an improvement for the VR-36 PCS score of 6.9 points (*p* = 0.06), very close to the aimed for 7 points. Additionally, there was improvement in SF McGill-2 total pain score and all five MFI subscales. The improvement in general and mental fatigue subscales of the MFI was of a greater degree than was seen in a study of Mind-Body Bridging intervention to improve sleep in GWI [97]. Of interest, we noted post hoc that 29% of the 21 participants who completed the 3-month follow-up no longer met Kansas case criteria for GWI. 

### 4.3. Adverse Events

While the Hubbard detoxification program is demanding in terms of time and effort, we did not detect any serious adverse events. The participant who developed moderate hypokalemia, which could have been managed with oral potassium, had a history of bowel resection, which increases risk for gastrointestinal adverse effects of supplements. There were no concerning changes in the clinical chemistry measures we monitored. 

### 4.4. Potential Mechanisms Underlying the Effects of the Regimen on Outcomes

We did not undertake an exploration in this study of what physiologic processes may have accounted for changes that we are reporting. However, some of the components are reasonably well understood. The Hubbard program was developed based on the hypothesis that the combination of exercise, induced sweating due to heat exposure from sauna and increasing doses of crystalline niacin will increase the rates at which the body will mobilize and excrete lipophilic and other xenobiotics [45,52]. Niacin (nicotinic acid, vitamin B3) is an essential micronutrient, with numerous functions including formation of nicotinamide adenine dinucleotide (NAD), as well as several other enzymes. NAD may be depleted when used as a substrate for repair of DNA strand breaks due to damage from xenobiotics [98]. Frank niacin deficiency leads to pellagra, characterized by dermatitis, gastrointestinal disturbances and cognitive changes, key symptoms seen in GWI, which may in part be due a state of intracellular or sub clinical pellagra as described in some health conditions [99] and being corrected with the niacin supplementation. Niacin alone may have antioxidant and anti-inflammatory effects [100], can protect or restore mitochondrial function [101], with mitochondrial DNA damage identified in GW veterans [34]. Niacin may decrease liver reactive oxygen species generation and inhibit production of the major pro-inflammatory cytokine, IL-8 [102]. Niacin status should be biochemically assessed in this population to detect possible sub-clinical deficiency and should be measured in a future clinical trial. 

Niacin also has long been used in humans to reduce serum lipid concentrations through direct effects on synthesis as well as a reduction of mobilization from adipose tissue stores [100,103,104]. However, it has been shown in both animals [105] and humans [106] that the response to niacin on free fatty acids is biphasic, with an initial suppression followed by an overshoot. This appears to be a direct effect on adipocytes, initially inhibiting lipolysis for one hour, then causing a dramatic increase in free fatty acid release from adipose tissue for over 24 hours [107]. Mobilization of lipid stores has been shown to be accompanied by a release of fat-stored toxins such as PCBs in both animal [108] and human [109] studies. Moderate aerobic exercise also mobilizes fat [110]. A combination of exercise and extended release niacin was shown to have synergistic effects on increasing PON1 concentration and activity, of interest due to the role of PON1 in hydrolyzing organophosphate pesticides and sarin [111] and as certain PON1 genotypes may be associated with GWI risk [31]. The Hubbard regimen also includes administration of supplemental oils which can further synergistically help overcome enterohepatic recirculation/reabsorption of lipophilic contaminants being excreted in bile [112]. 

Enhanced sweat production itself plays a major role in the regimen, as it coincides with the increased turnover of lipophilic xenobiotics and possibly hydrophilics immediately following the effects of both the niacin and moderate aerobic exercise. Sweat composition is complex, includes sebum, contains urea and medium molecular weight molecules such as cytokines and has its own metabolomics signature [113]. Sweating alone has been used to reduce uremia, and has long known similarities to urine composition, although concentrations differ. [93,114,115]. Lipophilic contaminants such as PCBs and dioxins, organochlorines and polybrominated flame retardants have been identified in sebum and numerous hydrophilic xenobiotics have been identified in sweat, including heavy metals and phthalates [51,116,117] as well as metabolites of or parent drugs of abuse [118,119,120,121]. Sweat volume can reach six liters in sauna sessions, an amount in excess of average daily renal output, so that some net loss of residual toxins or their metabolites via sweat and sebum is probable, given sufficient sweating and provided that these can be mobilized from tissue storage compartments. Overall, the combined steps of the regimen may shift disposition of xenobiotics from storage compartments and enhance their metabolism and excretion. 

Aerobic exercise alone (1 hour 3–4 times per week for 12 weeks) was found to have modestly improved fatigue, distress, cognitive symptoms and mental health functioning in a large 2003 GWI illness treatment trial [4]. While it is not likely that exercise alone would make significant impact on toxicant effects, it could itself be a component of symptom improvement. There is also the possibility that the benefit from this program is primarily psychological. Most of us would feel better if we relaxed and exercised, and sat in warm sauna for several hours per day and were constantly attended to by others. In addition, many people are on medications that are often not needed. However, if the benefit were only psychological, one would not expect it to last for three months. While there may be a psychological component, it seems unlikely that this accounts for all of the benefits observed.

### 4.5. Exposure Essessment

It has been 29 years since the first Gulf War, when the exposure hazard occurred during a discreet time period. Exposures were multiple and variable and no internal exposure assessments exist that would even allow conjecture as to what internal doses were and what if anything would still remain. However, there is clear evidence that some persistent organic pollutants remain in the human body with half-lives longer than this period of time, depending upon their structure and the concentration. For both PCBs and dioxins, it is known that rates of elimination are more rapid when concentrations are high than when they are lower. Hopf et al. [122] calculated that the half-life at low internal dose was 21.83 years for Aroclor 1242 and 133.33 years for Aroclor 1254. Both are commercial mixtures containing both lower chlorinated congeners that are more rapidly metabolized, and more resistant higher chlorinated congeners. Quinn and Wania [123] modeled parameters for various PCB congeners and calculated a half-life of 231 years for PCB 180, a congener with seven chlorines. The half-lives of organophosphates are generally briefer that those for PCBs, but it is still possible that some agent from the Gulf War is maintained for long periods of time in GWI veterans, and that the Hubbard detoxification protocol facilitates release. 

The current work aligns within the exposome paradigm, with the identified need for untargeted biomonitoring approaches to characterize as yet unknown low level xenobiotic exposures and relationships to adverse effects [124]. Consistent with this we already know that earlier deployments may also have contributed to the body burdens of some veterans. For instance, White et al. reported 17.8% of GW veterans had served in Vietnam where there was possible exposure to dioxin [125,126]. The lifetime exposome is a better way to understand our hypothesis that symptoms and functional improvement herein reported for this ‘detoxification’ approach are at least in part attributable to reduced body burden, even if of more persistent chemicals. Perhaps with Gulf War deployment, exposure exceeded a threshold of body tolerance for some personnel, and these were additive to earlier exposures to more persistent toxins [2].

### 4.6. Strengths and Limitations

There are a number of significant strengths in this study. The trial was pragmatic, delivering the intervention using an existing provider in the community, and the intervention itself has been in use for decades, without altering its components. Multiple approaches enhanced intervention fidelity, including a daily written report system and trained regimen supervisors following a detailed manual. We had complete follow-ups with no losses from the intervention or control group for the first intervention phase. The intervention itself was not costly, and as it is of relatively short duration (weeks) and does not need to be administered repeatedly or chronically.

The randomized control design with blinded allocation is a strength reducing selection bias and the waitlist changes can account for regression to the mean. Of note, the lack of improvement in symptoms and quality of life in veterans with GWI is well documented for over 25 years and can be used as an historical comparison making regression to the mean less likely. The waitlist control was of additional value as there are no studies of the natural variation of symptoms in a short term in these GW veterans, nor whether they vary in their Kansas GW case status. Overall, the veterans who volunteered for our study appeared to be representative of other GWI populations in the literature, regarding HRQoL, symptoms and cognitive impairments, improving the generalizability of our findings [127]. Our participants were also representative in terms of the number who reported comorbidities. 

There are limitations inherent in a pilot study with regard to conclusions regarding efficacy and only estimates can be made. Our sample size was small but in line with recommendations that 20 to 40 participants would be sufficient for pilot and feasibility studies [128], and not unlike other pilot studies [129]. However, our use of simple randomization resulted by chance in unequal balance with a smaller number in the waitlist group, which could be prevented by block randomization. Our 3-month follow-up period may be too short to predict long term outcomes. Selection bias could have been introduced as we could not accept veterans currently on analgesics or anti-depressants; thus, those with more severe pain or depression may have been selectively excluded. Alternately, our sample may comprise a group of more seriously ill GWI veterans highly motivated to find something that will relieve their symptoms. The required time for the study likely introduced a selection bias, as we could not get some consented participants to agree to continue intake steps once they fully understood the time commitment; this is a problem with waitlist designs, where waiting too long for an intervention may lead to control group attrition. 

However, we found the population selected to our study was similar to the GWI population in other studies, as discussed above, so selection bias should not greatly affect external validity, and as we used concealed allocation to randomize those selected, we should have reduced the effects of selection bias on internal validity. However, we had reduced retention for the 3-month follow-ups, limiting the generalizability over a longer term, as there may have been differences in symptoms and quality of life in those who did not attend. 

It was not possible to blind the participants or the staff at the facility, although outcome assessments were blinded. Although blinding of participant allocation does not seem feasible, an active control could be considered, such as daily short saunas and placebo supplements for 3 weeks without niacin. However, lack of niacin flush would reveal allocation. 

We relied on patient reported outcome measures, which are subjective, have problems with possible recall bias and have thresholds of discrimination that are difficult to accurately quantify [130]. We are reporting that roughly a third of the participants no longer met the Kansas case criteria for Gulf War illness at the 3 month follow-up; while widely used [6,70] and recommended for research by the Institute of Medicine [10], the validity of this case definition is still undetermined and its natural history of variation of scores among ill Gulf War veterans has not been studied. None of the current case definitions fully account for the clinical heterogeneity commonly seen in veterans with Gulf War Illness; for instance, some veterans with GWI do not have pain, while with others pain is severe. Thus, we may be misestimating some of the improvements we are reporting regarding which particular veterans might benefit, or how much [131]. The study did not include any objective biomarkers, and was not designed to test the hypothesis that the Hubbard detoxification program results in reduction of body burden of xenobiotics. However, we have a biobank of >100 serum samples from the participants at each time point for later analysis for persistent chemicals, should funding become available.

This pilot project has generated observations that can be utilized in a future definitive trial. A future RCT would need to highlight the requirement for several weeks of time commitment for participants, which is doubled for those in a waitlist control group, to avoid misunderstandings. It would be advisable to exclude veterans with Crohn’s disease or prior bowel resection due to the gastrointestinal effects of multiple supplements ingestion. Follow-ups should be made available by secure web access especially for participants who live at a distance from the facility. Follow-ups for period longer than 3 months would also be valuable.

Ideally, a future study would include biomarkers of GWI case status, such as serum autoantibodies [38], or other objective measures such as longer term follow-up changes in hippocampal microstructure [132]. Additionally, biomarkers should be obtained that would represent, in real time, the enhanced detoxification process such as changes in adipose, serum or skin lipids xenobiotic burden [51,52,133,134,135]. Another potential tool is measurement of concentrations of small molecular weight compounds in both serum and sweat before, during and after the regimen [136,137,138].

## 5. Conclusions

Our study provides preliminary evidence of sustained improvement in a number of measures of human health and function in GWI veterans after application of the Hubbard detoxification program. The intervention was safe and well-tolerated and the study was feasible with an acceptable recruitment rate and excellent retention through the sauna intervention. The estimated benefits to the GWI veterans in our study suggest potential value to the larger veteran population who may have other deployment exposures such as burn pits [139], and for civilians with other toxicant exposures. The intervention has the potential to provide benefit with minimal disruption of the lives of individuals. This aligns with conclusions from a recent review [3], “the identification of treatments for the GW veteran population will have far-reaching implications for treating other groups of ill patients for whom no effective treatments have been identified”, and as such, it provides a measure of hope which should be expanded to a larger RCT.

## Figures and Tables

**Figure 1 ijerph-16-04143-f001:**
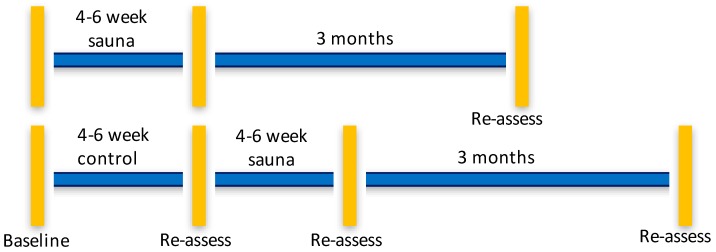
Timeline of assessment points for sauna intervention and waitlist control groups. Vertical bars indicate assessments of all outcomes were performed. Horizontal lines indicate the time frame of the study and follow-ups.

**Figure 2 ijerph-16-04143-f002:**
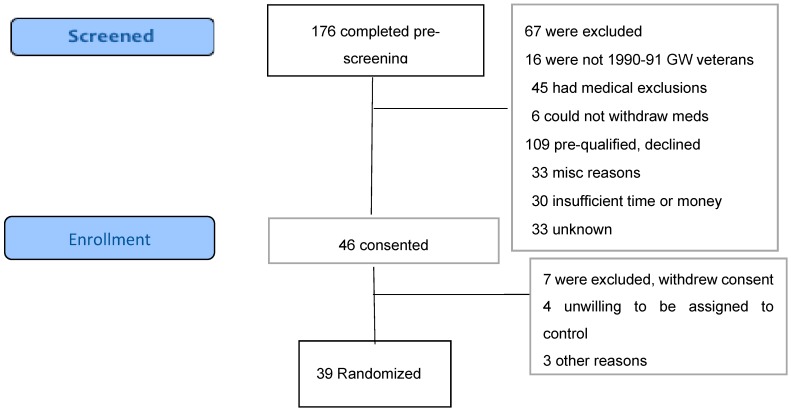
CONSORT Flow diagram.

**Table 1 ijerph-16-04143-t001:** Baseline and health characteristics of the study participants.

Characteristic	Intervention *n* = 22	Control *n* = 10
Age (years) mean ± SD	51.7 ± 7.9	50.2 ± 5
BMI (kg/m^2^) mean ± SD	32.3 ± 6.2	32.0 ± 6.3
Sex [*n* (%)]	
Male	14 (64)	7 (70)
Female	8 (36)	3 (30)
Race [*n* (%)]	
White	20 (91)	5 (50)
Black	1 (1)	5 (50)
Other	1 (1)	0 (0)
Married [*n* (%)]	15 (65)	4 (36)
Employment [*n* (%)]	
Fulltime	9 (41)	10 (100)
Part time, unemployed, retired	3 (13)	0 (0)
Disabled	10 (41)	0 (0)
Current smoker [*n* (%)]	4 (18)	0 (0)
Smoked during Gulf War [*n* (%)]	3 (14)	3 (30)
Comorbidities [*n* (%)]	
# diabetes (DM)	3 (13)	2 (15)
# who remained on medications (DM, blood pressure, Other)	11 (48)	5 (45)
# who recently stopped antidepressants	11 (48)	1 (8)
# who recently stopped analgesics	8 (35)	2 (6)
Symptom and function assessments- mean ± SD	
VR-36 physical component summary	30.3 ± 8.9	36.7 ± 10.5
VR-36 mental component summary	38.4 ± 12.1	40.5 ± 14.1
physical functioning	43.8 ± 20	59.4 ± 28.8
role-physical	35.6 ± 29.3	54.6 ± 37.2
bodily pain	32 ± 26.2	44.0 ± 29.1
general health	27.3 ± 18.5	40.9 ± 27.3
vitality	23.3 ± 23.1	32.0 ± 25.3
social functioning	36.9 ± 27.7	46.6 ± 35.0
role-emotional	53 ± 31.3	61.7 ± 34.5
mental health	53 ± 22.2	58.0 ± 24.2
Multidimensional fatigue inventory	
general fatigue	16.9 ± 3.2	15.3 ± 5.3
physical fatigue	16.1 ± 4.0	12.6 ± 5.7
reduced activity	15 ± 4.3	12.4 ± 4.7
reduced motivation	13.7 ± 4.0	11.8 ± 3.8
mental fatigue	14.6 ± 2.8	15.4 ± 4.6
SF McGill Pain Q-2 total pain	4.2 ± 1.8	3.9 ± 2.3
Kansas GWI case criteria (number positive of 6)	4.5 ± 1.2	4.6 ± 0.7

Abbreviations: SD = standard deviation, BMI = body mass index, VR-36 = Veterans RAND 36-Item Health Survey, SF McGill Pain Q-2 = Short form McGill Pain questionnaire version 2, # = number.

**Table 2 ijerph-16-04143-t002:** Effect of treatment allocation on the health measures outcomes at 6 weeks while controlling for measures at baseline *.

Health Measures	Waitlist (WL) Control (Usual Care) *n* = 10		Intervention *n* = 22		Adjusted between Group Differences^1^ Comparing Scores between WL and Intervention at Week 6	*p*-Value
VR−36 quality of life	Baseline	6-wk follow-up	Baseline	6-wk follow-up	(95% confidence interval)	
VR-36 physical component summary	36.7 (10.4)	35.7(12.10)	30.3 (8.9)	38.2 (10.3)	6.9 (−0.3, 14.2)	0.06
VR-36 mental component summary	40.5 (14.1)	41.1 (12.8)	38.4 (12.1)	49.2 (9.3)	9.5 (3.1, 15.8)	0.003
physical functioning	59.4 (28.8)	63.5 (30.4)	43.8 (20.0)	59.5 (26.7)	2.7 (−18.1, 23.5)	0.8
role-physical	54.6 (37.2)	45.0 (36.9)	35.6 (29.3)	61.9 (29.9)]	27.6 (6.9, 48.3)	0.009
bodily pain	44.0 (29.1)	36.0 (28.0)	32.1 (26.2)	56.2 (27.0)	26.4 (8.5, 44.4)	0.004
general health	40.9 (27.3)	38.5 (31.1)	27.3 (18.5)	45.7 (24.2)	20.7 (9.2, 32.3)	< 0.001
vitality	32.0 (25.3)	30.0 (28.6)	23.3 (23.1)	53.9 (25.2)	31.2 (15.6, 46.9)	< 0.001
social functioning	46.6 (35.0)	52.5 (32.2)′	36.9 (27.7)	64.2 (26.5)	15.9 (−3.9, 35.7)	0.1
role-emotional	61.7 (34.5)	59.2 (41.1)	53.0 (31.3)	68.9 (26.1)	15.2 (−4.9, 35.2)	0.1
mental health	58.0 (24.2)	59.6 (25.6)	53.0 (22.2)	73.8 (18.4)	17.7 (5.3, 30.0)	0.005
Multidimensional Fatigue Inventory						
general fatigue	15.3 (5.3)	15.7 (3.6)	17.4 (2.6)	12.8 (4.8)	−4.3 (−7.4, −1.3)	0.006
physical fatigue	12.5 (5.7)	14.6 (4.8)	16.2 (4.0)	13.1 (4.7)	−3.5 (−6.9, −0.2)	0.04
reduced activity	12.4 (4.7)	15.4 (4.9)	15.2 (4.3)	12.8 (4.4)	−4.0 (−7.3, −0.7)	0.02
reduced motivation	11.8 (3.8)	12.2 (3.2)	13.7 (4.0)	10.2 (3.9)	−3.1 (−5.6, −0.5)	0.02
mental fatigue	15.4 (4.6)	16.2 (4.5)	14.6 (2.8)	10.4 (4.0)	−5.7 (−8.7, −2.7)	< 0.001
SF McGill Pain Questionnaire-2						
Total pain score	3.9 (2.5)	3.0 (2.0)	4.2 (1.8)	2.1 (1.5)	−1.1 (−2.0, −0.2)	0.02
Kansas GWI case criteria						
Total of six domains score	4.7 (0.7)	3.3 (1.8)	4.5 (5.3)	2.7 (3.7)	−0.5 (−1.9, 0.9)	0.5
Proportion positive	10/10 (100%)	8/10 (80%)	22/22 (100%)	11/22 (50%)		

Abbreviations: VR-36 = Veterans RAND 36-Item Health Survey, SF McGill Pain Q-2 = Short form McGill Pain questionnaire version 2, ANCOVA = analysis of covariance. * Data are expressed as estimates and 95% confidence intervals adjusted for baseline values. Positive changes for the VR-36 S scores indicate improvement while negative changes for the SF McGill Pain Questionnaire-2 and Multidimensional Fatigue Inventory indicate improvement.

**Table 3 ijerph-16-04143-t003:** Estimated Magnitude of Change Over Time to 3 months By Group ^#^.

Health Measures	Waitlisted	Waitlisted	Intervention	Intervention
Parameter	Week 6 ^§^ vs. Baseline *	Month 3 vs. Week 6	Week 6 vs. Baseline	Month 3 vs. Week 6
VR-36 quality of life			
VR-36 PCS	5.8 (0.45, 11)	−1.3 (−8.2, 5.7)	7.8 (4.2, 11.5)	−1.3 (−5.3, 2.7)
VR-36 MCS	2.2 (−3.9, 8.4)	7.3 (−0.7, 15.3)	11.1 (6.7, 15.3)	−0.1 (−4.8, 4.5)
VR-36 Sub scores			
Physical functioning	11 (−3.9, 25.9)	−4.9 (−24.1, 14.4)	15.6 (5.5, 25.7)	−0.2 (−11.5, 11)
Role−physical	15 (−1.1, 31.1)	2.1 (−18.7, 23.0)	26.7 (15.7, 37.6)	−6.5 (−18.6, 5.7)
Bodily pain	7.5 (−8.1, 23.8)	2.4 (−18.7, 23.6)	22.9 (11.8, 34.1)	−7.9 (−20.3, 4.4)
General health	16.6 (5.9, 27.4)	1.4 (−12.7, 15.5)	18.2 (10.8, 25.5)	−2.7 (−10.9, 5.5)
Vitality	11.5 (−3.1, 26.1)	29.6 (10.7, 48.5)	31.4 (21.5, 41.4)	13.3 (2.3, 24.3)
Social functioning	6.2 (−10.5, 23)	3 (−18.7, 24.7)	27.6 (16.2, 39)	−6.5 (−19.1, 6.2)
Role−emotional	9.2 (−8.4, 26.7)	4.1 (−18.7, 26.9)	15.8 (3.9, 27.8)	2.6 (−10.7, 16)
Mental health	4 (−7.2, 15.2)	10.4 (−4.1, 25)	21.3 (13.7, 28.9)	−6.6 (−15, 1.9)
Multidimensional fatigue inventory			
General fatigue	−2.8 (−5.5, −0.1)	−1.6 (−4.7, 1.5)	−4.1 (−5.9, −2.4)	1.2 (−0.8, 3.1)
Physical fatigue	−3 (5.9, − 0.1)	−1.1 (−4.5, 2.2)	−3 (−4.9, −1.2)	0.5 (−1.6, 2.6)
Reduced activity	−3.6 (−6.9, −0.2)	−3.6 (−7.5, 0.3)	−2.3 (−4.5, −0.2)	−0.4 (−2.8, 2)
Reduced motivation	−2.7 (−5.2, −0.2)	−1.3 (−4.2, 1.6)	−3.2 (−4.8, −1.6)	0.6 (−1.2, 2.4)
Mental fatigue	−2.3 (−5.2, 0.5)	−3.9 (−7.2, −0.7)	−4.1 (−5.9, −2.4)	1.7 (−0.3, 3.7)
SF McGill Pain Questionnaire-2			
Total pain score	−1 (−1.9, −0.1)	0.6 (−0.5, 1.7)	−2.2 (−2.7, −1.6)	0.3 (−0.3, 1.0)

Abbreviations: VR-36 = Veterans RAND 36-Item Health Survey, PCS = Physical Component Summary score, MCS = Mental Component Summary Score. ^#^ Estimated Effect (95% CI), linear mixed model. ^§^ Week 6 refers to the measures obtained after each group had received the intervention. * Baseline refers to the second baseline measures for the waitlisted group, after having only usual care and just before they entered the sauna. Positive changes for the VR-36 S scores indicate improvement, while negative changes for the SF McGill Pain Questionnaire-2 and Multidimensional Fatigue Inventory indicate improvement.

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
