# Peer review of "A Detoxification Intervention for Gulf War Illness: A Pilot Randomized Controlled Trial"

_ijerph, 2019, doi:10.3390/ijerph16214143_

Round 1

Reviewer 1 Report

Dear Author(s),

Thanks for submitting such an interesting paper. Please find my comments below:

Extensive and critical evaluation of previous work  Excellent rationale for research methodology and data collection methods. It is advised to include the critical awareness of strengths and weaknesses of the approach Logical and coherent discussion, very clearly articulated and developed. Insightful, well-supported conclusions that flow logically from the work presented. Line 99-102 requires referencing Fig 1 is blur and the legend of figures needs to be written under the figure with a full description.

Thanks

Author Response

Reviewer 1: 

Thanks for submitting such an interesting paper. Please find my comments below:

Extensive and critical evaluation of previous work. Excellent rationale for research methodology and data collection methods.

Thank you so much.

It is advised to include the critical awareness of strengths and weaknesses of the approach. We have revised our discussion of these points in 2.1 Study design section. This was also covered more fully in Section 4.6.

Logical and coherent discussion, very clearly articulated and developed. Insightful, well-supported conclusions that flow logically from the work presented.

Line 99-102 requires referencing

This was provided, now on line 125.

Fig 1 is blur and the legend of figures needs to be written under the figure with a full description.

We have replaced Fig 1 with a revised clearer version with expanded legends for figures and tables.

Reviewer 2 Report

The trial design contains several flaws that affect the interpretation of study results:

1) The trial's power is extremely low, even for a pilot study. This is particularly concerning given how inherently diverse the target population seems to be.

2) There is no concurrent control group. Both groups get the intervention, and, as the authors pointed out, it is impossible to blind either the study participants or the clinicians to whom is in each group. In this study, it could be said that each subject acts as his/her own control, since symptomology and clinical pathology were analyzed before and after in all participants. However, it is unclear how representative the study population was of the GWI-population as a whole. Another control group comprised of GWI-sufferers that weren't given any intervention at all for the entire length of the trial would be a more appropriate control.

3) As the trial participants were self-selected and the trial outcome measure (symptomology) was self-reported, there is a strong possibility of bias in both study population selection and outcome reporting.

4) The purpose of the study as stated in the manuscript was to test the effectiveness of a regimen intended to 'detoxify' the body in remediating symptoms of GWI. However, the study design does not explicitly measure whether the intervention actually achieves significant removal of posited causative agents of GWI, such as PCBs, from the trial participants.

Author Response

Reviewer 2:

The trial design contains several flaws that affect the interpretation of study results:

1) The trial's power is extremely low, even for a pilot study. This is particularly concerning given how inherently diverse the target population seems to be.

As stated in our manuscript line 537, “As this is a pilot study, a formal sample size calculation was not required”, based on the CONSORT guidelines for Pilot and Feasibility studies and per Lee et al. BMC Medical Research Methodology 2014,Pilot studies are not formally powered to assess effect.” With regard to sample size, our total of 22 in the intervention arm and 10 in the control arm, while unequal, is within the ranges found in a recent audit of pilot study sample sizes (Billingham et al. 2013).

To clarify we judged that our sample size would be adequate to assess overall feasibility of recruitment and we have revised the manuscript to indicate that only estimates of effect were being sought on line 540.

We agree that veterans with GWI are an heterogeneous group, however all our participants did meet the Kansas case criteria, which was recommended for GWI research by the Institute of Medicine. To account for the diversity in our sample population we used ANCOVA adjusted for baseline values in estimating changes in outcomes.

2) There is no concurrent control group.

There was in fact a concurrent control group (the wailtlist group), as this was a randomized controlled trial. See the first set of data we present in Table 2, which label has been modified to “Table 2. Effect of treatment allocation on health measures outcomes at 6 weeks while controlling for measures at baseline.”

Both groups get the intervention, and, as the authors pointed out, it is impossible to blind either the study participants or the clinicians to whom is in each group.

In the first phase of the trial which is the RCT, the control group did not get the intervention. By the time of 3 month follow up, yes both groups had had the intervention. Our allocation was blinded, as was the data analysis of outcomes.

In this study, it could be said that each subject acts as his/her own control, since symptomology and clinical pathology were analyzed before and after in all participants.

We predicted or hypothesized that the intervention would have long-term effects thus it would not be possible to do a crossover trial, as the immediate intervention group would not be expected to ‘washout’ the effects of the intervention. Thus, we could not use each subject as their own control.

The data on the control group’s post intervention outcomes was presented in Table 3, line 743,(which has had revisions of a legend and improved title) where the control group post intervention outcomes were used as a second baseline to compare to their 3 month outcomes, essentially to assess for stability over time and to see if doing the intervention after a waitlist made any difference between groups in the 3 month outcomes which it did not.

However, it is unclear how representative the study population was of the GWI-population as a whole.

We did consider representativeness in lines 592. “The PCS and MCS and individual domain scores in our study were very similar to scores in other studies of GWVs [4, 55, 82] indicating that our sample was likely representative of this population.”  Also discussed in lines 990, Overall, the veterans who volunteered for our study appeared to be representative of other GWI populations in the literature, regarding HRQoL, symptoms and cognitive impairments, improving the generalizability of our findings [133]. Our participants were also representative in terms of the number who reported co-morbidities.”

But we agree there was heterogeneity, particularly in terms of GW vets with considerable pain and those with little pain. However, we controlled for baseline pain in the ANCOVA.

Another control group comprised of GWI-sufferers that weren't given any intervention at all for the entire length of the trial would be a more appropriate control.

Agreed, and this would be the best design for a larger trial, as long as the veterans would agree to this.

3) As the trial participants were self-selected and the trial outcome measure (symptomology) was self-reported, there is a strong possibility of bias in both study population selection and outcome reporting.

We agree that some selection bias was an unavoidable limitation, as was discussed, line 1003, Selection bias could have been introduced as we could not accept veterans currently on analgesics or anti-depressants, thus those with more severe pain or depression may have been selectively excluded. Alternately, our sample may comprise a group of more seriously ill GWI veterans highly motivated to find something that will relieve their symptoms. The required time for the study likely introduced a selection bias, as we could not get some consented participants to agree to undergo randomization and continue intake steps, once they fully understood their possible time commitment, a problem with waitlist designs where waiting too long for an intervention may lead to control group attrition.” We added the above sentence to the manuscript, line 1016. However, we found the population selected to our study was similar to the GWI population in other studies, as discussed above, so selection bias should not greatly affect external validity, and as we used concealed allocation to randomize those selected, we should have reduced the effects of selection bias on internal validity.

As regards our use of symptoms and function questionnaires as outcome measures, we selected well validated and commonly used patient reported outcome instruments, used in many other GW studies, also selected by the recent NINDS CDE initiative (https://cdmrp.army.mil/gwirp/pdfs/GWI%20CDE%20Draft%20Version%201_0.pdf) and discussed the existing literature regarding their use in accordance with CONSORT PRO guidelines. Limitations re these measures were further revised line 1032.

4) The purpose of the study as stated in the manuscript was to test the effectiveness of a regimen intended to 'detoxify' the body in remediating symptoms of GWI. However, the study design does not explicitly measure whether the intervention actually achieves significant removal of posited causative agents of GWI, such as PCBs, from the trial participants.

Our purpose at this stage was solely to determine if the regimen improved symptoms, see lines 825-829. “The study did not include any objective biomarkers, and was not designed to test the hypothesis that the Hubbard detoxification program results in reduction of body burden of xenobiotics.  However, we have a biobank of >100 serum samples from the participants at each time point for later analysis for persistent chemicals, should funding become available.”

Reviewer 3 Report

The Introduction is the authors’ take on GW exposure studies and could be shortened and more focused. Whether the length and detail is of interest to this journal’s readers is up to the Editor. What is lacking, as is the case in many other such reviews, are pathogenesis-grounded hypotheses as to why most of these exposures (one-time period, nearly 30 years ago) should still be contributing to clearly worsening health across this cohort. Absent this, the reader is left to conclude that the cited exposures cause Gulf War Illness, a speculative contention. There are other more solidly grounded perspectives on GWI causation (i.e., Tsilibary et al. (2018) Journal of Neurology and Neuromedicine. 3(5): 23-28) that the authors need to consider. There are more recent reports of physical exercise, mindfulness, and other interventions with GWI that should be referenced. Nancy Klimas has published on CoQ10 as an intervention for GWI. These are arguably more relevant to the present study than those currently cited. Page 2 line 87 starts with an incomplete sentence. Engdahl [28] did not report altered brain structure, but altered brain function in GWI. Is it a disservice to the reader to refer to neuropsychological results as being “in a separate publication” (line 181)? If it is not a disservice, where might one find the other publication? It is questionable to use another GW researcher’s change scores (line 405) to define “clinically meaningful”. Why not cast the discussion in terms of “effect size” or change in SD, as the authors did with the MPQ? The authors appeal to IND standards as applicable to this study, then on line 208, suggest that this is part of an IND application. On Line 358, it is stated that the IND approval was in force. Did it specifically include such items as up to 5000 mg niacin? References for the Hubbard method are all positive. How do the authors reconcile their justification for it with the substantial refutations it has experienced when reviewed by others? https://en.wikipedia.org/wiki/Purification_Rundown Line 674 suggests this study supports the value of the Hubbard detoxification approach when in fact, the “active ingredient” (if any) remains to be identified. Is it the exercise, supplements, sauna, or continuous attention from caring providers? The authors assert that the method confirms that toxicants are somehow removed from adipose tissue through this method, a controversial and unsupported claim. Blood samples will be assayed in the future? The stored blood samples will need to be analyzed for toxicant levels to support this claim.

Author Response

Reviewers 3:

The Introduction is the authors’ take on GW exposure studies and could be shortened and more focused. Whether the length and detail is of interest to this journal’s readers is up to the Editor.

We hope this will be of interest.

What is lacking, as is the case in many other such reviews, are pathogenesis-grounded hypotheses as to why most of these exposures (one-time period, nearly 30 years ago) should still be contributing to clearly worsening health across this cohort. To the best of our knowledge an accepted pathogenesis-grounded hypothesis for GWI is still being sought. As recently summated in DNA Cell Biol. 2019 Jun;38(6):561-571. doi: 10.1089/dna.2018.4469,  GWI is a complex disorder with a likely combination of both genetic and environmental components, culminating into altered epigenetic status.

Absent this, the reader is left to conclude that the cited exposures cause Gulf War Illness, a speculative contention.

We hoped to prevent such speculation in lines 123-127. We are making no claims that veterans with GWI have abnormally high levels of specific toxins, only that the unusual set of GW deployment exposures (along with genetics and other factors) are likely linked with precipitating the illness. Humans are continuously exposed to complex mixtures of xenobiotics, which for these veterans would have occurred prior to, during and following the Gulf war exposures, all contributing to accumulated body burden.”

Also see lines 100-103, where while there are many confounders, the continuing symptoms may be hypothesized to be due to: persistence of some chemical agent or metabolite in the body, permanent damage to the brain and/or immune system, genotoxic mitochondrial damage or some combination of persistent exposure and tissue damage.” We have now added reference to persistent antigens, as you cite in Tsilibary, below.

There are other more solidly grounded perspectives on GWI causation (i.e., Tsilibary et al. (2018) Journal of Neurology and Neuromedicine. 3(5): 23-28) that the authors need to consider.

This line of research regarding HLA class 2 alleles is truly exciting and we follow it with great interest, especially in terms of future clinical trials. But the exposures, if only vaccines, obviously were necessary for the effect.

There are more recent reports of physical exercise, mindfulness, and other interventions with GWI that should be referenced. Nancy Klimas has published on CoQ10 as an intervention for GWI. These are arguably more relevant to the present study than those currently cited.

We are not aware of recent reports on exercise for GWI being of any major benefit, in fact exercise may trigger symptom exacerbation, e.g. Garner et al ( Am J Transl Res. 2018) but we had included the report of some benefit from exercise from Donta in 2003, on line 927-9. We are aware of the study by Beatrice Golomb of CoQ10 and added this reference this to the paper, but have not yet seen the results of Nancy Klimas’s ongoing trial with CoQ10. We also added at your suggestion, references to carnosine, mindfulness and acupuncture, line 61.

Page 2 line 87 starts with an incomplete sentence. 

The sentence was made clearer.

Engdahl [28] did not report altered brain structure, but altered brain function in GWI.

Thank you for noting this, the text has been corrected. (line 136)

Is it a disservice to the reader to refer to neuropsychological results as being “in a separate publication” (line 181)? If it is not a disservice, where might one find the other publication?

Due to the length of the paper, we decided to split our results into 2 papers. The neuropsychological manuscript is in preparation, and will refer to the methods section of the current paper. We are sorry we cannot provide more information as yet.

It is questionable to use another GW researcher’s change scores (line 405) to define “clinically meaningful”. Why not cast the discussion in terms of “effect size” or change in SD, as the authors did with the MPQ?

Thank you for noting our inconsistency and we have removed the reference to 0.5 SD for the MPQ as we are following CONSORT recommendations, which refer to “effect size” in its generic sense. The CONSORT statement glossary describes the term effect size as equivalent to “treatment effect,” which is explained as “Commonly expressed as …difference in means for continuous outcomes.” and use of the other meanings of effect size would not be recommended for pilot study.

We also clarified our use of another GW researcher’s change scores (line 405) to define “clinically meaningful” which in our view is not questionable, and we hope the addition to the sentence on line 498-500 will make this clearer as this may not have been fully expressed. This whole topic of determining what would be clinically meaningful is not cut and dried, and we were not aiming to “define” this but to give some comparison. This is also in accordance with CONSORT PRO, “Further interpretation of PRO results may include discussion of a minimal important change…or... comparison with other similar RCTs.”

The authors appeal to IND standards as applicable to this study, then on line 208, suggest that this is part of an IND application. On Line 358, it is stated that the IND approval was in force. Did it specifically include such items as up to 5000 mg niacin?

We clarified the wording re the IND, line 267-8, as yes, the IND approval was in force. All components of the regimen including niacin dosing to 5000 mg, were included the FDA application and approval. The dosing schedule is further described in supplemental appendix, Investigator’s Brochure.

References for the Hubbard method are all positive. How do the authors reconcile their justification for it with the substantial refutations it has experienced when reviewed by others? https://en.wikipedia.org/wiki/Purification_Rundown

We did reference a negative publication (case report), line 470-471, reference [58], and have added more details regarding this to the text in section 2.7 Safety aspects, of a man who developed hyponatremia from overhydration, which prompted our precautionary approach to monitoring. We are surprised at this reviewer’s reliance on Wikipedia which is known to publish unverified data and opinions. (Bould et al, References that anyone can edit: review of Wikipedia citations in peer reviewed health science literature, BMJ. 2014.)

Line 674 suggests this study supports the value of the Hubbard detoxification approach when in fact, the “active ingredient” (if any) remains to be identified.

It is our impression that the regimen components are synergistic. See lines 866-870, “The Hubbard program was developed based on the hypothesis that the combination of exercise, induced sweating due to heat exposure from sauna and increasing doses of crystalline niacin will increase the rates at which the body will mobilize and excrete lipophilic and other xenobiotics.”

We agree that the “active ingredient” (if any) remains to be identified, and expect it will remain that way, as it is more likely the combined additive effects of each ingredient.

Is it the exercise, supplements, sauna, or continuous attention from caring providers? 

See lines 920-929 where this question was discussed. ”While it is not likely that exercise alone would make significant impact on toxicant effects it could itself be a component of symptom improvement. There is also the possibility that the benefit from this program is primarily psychological.  Most of us would feel better if we relaxed and exercised, and sat in warm sauna for several hours per day and were attended to by others. In addition, many people are on medications that often are not needed. However, if the benefit were only psychological one would not expect it to last for three months.  While there may be a psychological component, it seems unlikely that this accounts for all of the benefits observed.”

As above it is assumed the program steps are inter related and synergistic.  Many veterans are or have taken multivitamins and/or had saunas in clubs or gyms, but these habits should be more carefully accounted for in a future trial.

The authors assert that the method confirms that toxicants are somehow removed from adipose tissue through this method, a controversial and unsupported claim. Blood samples will be assayed in the future? The stored blood samples will need to be analyzed for toxicant levels to support this claim.

Agreed. We are preparing a proposal to proceed with this step.

Earlier trials conducted measurements in adipose tissue which showed reductions in PCBs (Schnare 1984 and Tretjak 1990 if you are interested).

Round 2

Reviewer 2 Report

No additional comments

Author Response

This reviewer had no comments other than that we do another spell check.  This has been done and some minor errors were corrected. 

Reviewer 3 Report

Line 465: “Heath” -> Health Reference 78 does not refer to test-retest validity, but to test-retest reliability. Table 1 should be reviewed for typos. Is it possible to get a VR-36 social functioning score as low as 11? (46.6+/-35.0)? What is the significance of standard parentheses vs. square parentheses? Table 3: The Waitlisted Month 3 vs. Week 6 VR-36 MCS Vitality score appears to have increased 29.6 points and the authors comment beginning on line 819. I was looking for this in section 3.4.2. The reader would need to see lab-based evidence that pre-treatment niacin levels were low before considering the possibly far-fetched hypothesis (sub-clinical pellagra) put forth on line 850 (this was hinted at beginning on the end of line 857). Section 4.4’s title (“examination of the mechanisms…”) is misleadingly titled as there is no true examination of mechanisms in the present paper. What follows are speculations about the mechanisms of the Hubbard method. This section should be re-titled and greatly shortened or eliminated. The proposition (line 903) that many were on unneeded medications is speculative and raises questions about the details of drug discontinuation sketched earlier in the paper. If this assertion is to be retained, specific examples need to be provided and confirmation that discontinued drugs were not renewed and were truly no longer needed is required. The supposition (line 904) that psychological benefits would not be expected to last 3 months is quite unfounded and needs to be deleted. Exposure assessment: the section on PCBs is irrelevant and should be deleted, as elevated PCBs have not been found in GW vets, as the authors previously noted. Line 949: “The intervention was not costly” is open to debate. Staff time, facilities cost, the income forgone by the (employed) veterans, travel and lodging costs are some of the cost factors to consider and would no doubt be high by most people’s standards. There is no bases for stating that the intervention does not need to be repeated (to maintain any gains), as the authors do not have long-term follow-up data. This should be deleted. This also applies to line 1036.

Author Response

Line 465: “Heath” -> Health Reference 78 does not refer to test-retest validity, but to test-retest reliability. Thanks, have corrected that.

Table 1 should be reviewed for typos. see below.

Is it possible to get a VR-36 social functioning score as low as 11? (46.6+/-35.0)? Yes, it is possible, and not rare. Have rechecked and two control participants and six immediate group participants were at the floor (ie 0) on the questions relating to social functioning, resulting in mean scores well below 1 SD.

What is the significance of standard parentheses vs. square parentheses?

Thanks for noticing. should have been standard parentheses, the square parentheses in Table 1 were a corruption, and have been corrected.

Table 3: The Waitlisted Month 3 vs. Week 6 VR-36 MCS Vitality score appears to have increased 29.6 points and the authors comment beginning on line 819. I was looking for this in section 3.4.2. We added a comment on the vitality sub score in section 3.4.2.  (This would not be “MCS Vitality”, but just “Vitality” as the MCS is a summary measure. In case the table was confusing we added a row below MCS entitled VR-36 sub-scores.)

The reader would need to see lab-based evidence that pre-treatment niacin levels were low before considering the possibly far-fetched hypothesis (sub-clinical pellagra) put forth on line 850 (this was hinted at beginning on the end of line 857). We agree but we did not have resources for these biochemical measurements (e.g. niacin number). We have modified the wording on line 870-1. Sub-clinical pellagra has been studied in various populations, (see Adv Food Nutr Res. 2018 Niacin. Kirkland Meyer-Ficca), one cause among others that might apply to GWVs being DNA damage, so we do not agree this is ‘possibly far-fetched’ as DNA damage consumes available NAD.

Section 4.4’s title (“examination of the mechanisms…”) is misleadingly titled as there is no true examination of mechanisms in the present paper. What follows are speculations about the mechanisms of the Hubbard method. This section should be re-titled and greatly shortened or eliminated. Good point, we re-titled the section. Although we did not ‘examine the mechanism’ per se, we think it is important to outline the biological plausibility of the components of the regimen.

The proposition (line 903) that many were on unneeded medications is speculative and raises questions about the details of drug discontinuation sketched earlier in the paper. If this assertion is to be retained, specific examples need to be provided and confirmation that discontinued drugs were not renewed and were truly no longer needed is required. GWI veterans were excluded if it “medically not advised to temporarily discontinue certain medications (e.g. anti-hyperlipidemics, antidepressants, analgesics and anti-inflammatories) for the period of the regime”. This was a decision between the participant and his/her own personal physician and was undertaken before arriving at the facility. They gave fully informed consent on this point.  It would be ideal to try to confirm discontinuation, but this would be a plan for another study and we felt the participants were highly motivated and understood why they would not take such meds. We did list the numbers of veterans who had discontinued the classes of such drugs in Table 1. The question of unneeded medications in veterans and any of us is under considerable review nowadays with new guides on safe deprescribing and avoiding polypharmacy. See also https://www.blogs.va.gov/VAntage/53716/helping-veterans-stop-unnecessary-medications/

The supposition (line 904) that psychological benefits would not be expected to last 3 months is quite unfounded and needs to be deleted. The psychological benefit we refer to would be similar to being on a holiday in a nice caring spa setting, which by common observation rarely gives lasting benefit.

Exposure assessment: the section on PCBs is irrelevant and should be deleted, as elevated PCBs have not been found in GW vets, as the authors previously noted. Nor were any GW vets tested for PCBs, so on that basis none were found. We felt however that some exposure to PCBs by troops deployed to the Gulf could not be ruled out due to the sediment measurements and body burdens of Iraqi’s. The question is not irrelevant as PCBs form a universal proportion of the human body burden as part of the exposome.

Line 949: “The intervention was not costly” is open to debate. Staff time, facilities cost, the income forgone by the (employed) veterans, travel and lodging costs are some of the cost factors to consider and would no doubt be high by most people’s standards. Agreed that there were time limited intervention related costs, but compared to costly medications needed for an indefinite period of time and other illness effects if remediated to some degree in the participants, it would not be seen as costly. A formal cost analysis could be done in the future if appropriate.

There is no bases for stating that the intervention does not need to be repeated (to maintain any gains), as the authors do not have long-term follow-up data. This should be deleted. This also applies to line 1036. Yes, we agree we cannot assert this and have deleted the phrase.